# When and How Does Mutation-Generated Variation Promote the Evolution of Cooperation?

**Mathias Spichtig and Martijn Egas \***

Institute for Biodiversity and Ecosystem Dynamics, University of Amsterdam, P.O. Box 94240,
1090GE Amsterdam, The Netherlands
**\*** Correspondence: egas@uva.nl; Tel.: +31-20-525-7748

**Abstract:** Mutation-generated variation in behavior is thought to promote the evolution of cooperation. Here, we study this by distinguishing two effects of mutation in evolutionary games of the finitely repeated Prisoner's Dilemma in infinite asexual populations. First, we show how cooperation can evolve through the direct effect of mutation, i.e., the fitness impact that individuals experience from interactions with mutants before selection acts upon these mutants. Whereas this direct effect suffices to explain earlier findings, we question its generality because mutational variation usually generates the highest direct fitness impact on unconditional defectors (*AllD*). We identify special conditions (e.g., intermediate mutation rates) for which cooperation can be favored by an indirect effect of mutation, i.e., the fitness impact that individuals experience from interactions with descendants of mutants. Simulations confirm that *AllD*-dominated populations can be invaded by cooperative strategies despite the positive direct effect of mutation on *AllD*. Thus, here the indirect effect of mutation drives the evolution of cooperation. The higher level of cooperation, however, is not achieved by individuals triggering reciprocity ('genuine cooperation'), but by individuals exploiting the willingness of others to cooperate ('exploitative cooperation'). Our distinction between direct and indirect effects of mutation provides a new perspective on how mutation-generated variation alters frequency-dependent selection.

**Keywords:** altruism; evolutionary game theory; frequency-dependent selection; mutation regime; Prisoner's Dilemma

## 1. Introduction

Evolution is driven by selection acting on heritable phenotypic variation. The amount of phenotypic variation can be described as a function of selection modulating the variation generated by mutation, recombination, and environmental factors. If selection is frequency-dependent, it can in turn be described as a function of the distribution of phenotypic variants in the population. A standard approach in evolutionary ecology is to ignore selection effects of phenotypic variation by reducing the analysis to interaction dynamics of an invader phenotype in an otherwise homogeneous resident population [1]. This approach is defensible if the addition of other phenotypic variants constitutes no more than noise on the selective interactions between these two phenotypes. Under a regime of frequency-dependent selection, however, increasing phenotypic variation may alter the outcome of evolutionary analyses qualitatively [2]. For example, increasing (mutation-generated) variation in behavior can promote the evolution of cooperation [2–7].

Studies of the evolution of cooperation typically document behavioral variation (e.g., [8–15]), including those where behavioral strategies are explicitly assessed (e.g., [16,17]). Hence, the existence of such behavioral variation may provide a general explanation for the evolution of cooperation [3–6]. However, this prediction is generated from studies simulating asexual populations, where each of

these studies is based on a particular choice of mutation regime, i.e., a mutation rate ($\mu$), a set of strategies that mutation can generate, and a rule of how mutant strategies tend to differ from the parental strategies (variation from recombination and from environmental factors is ignored). Given the relative lack of knowledge of the genetic underpinning of behavioral traits, arbitrary choices are inevitable. In asexual organisms, mutation is typically seen as the main source of heritable variation, and the mutation regime determines the expressed (heritable) variation. Yet, it is unclear whether the effect of mutation-generated variation in behavior to promote the evolution of cooperation is general, i.e., holds under any mutation regime. Therefore, we investigate whether variation-promoted evolution of cooperation is robust to the choice of mutation regime.

To analyze the effects of mutation regime on the evolutionary dynamics, we distinguish direct and indirect effects of mutation by using a narrow-sense definition of mutants: mutants are the individuals with a genotype distinct from their parent, i.e., a mutation event occurred during the generation of the individual (transforming its genotype). As a consequence of this narrow-sense definition, faithfully replicated offspring of mutants are not mutants themselves (in contrast to the use of the term in "mutant-wild type" contexts where faithfully replicated offspring of mutants remain categorized as the "mutant type"). Because selection acts on variation after it is generated by mutation, mutants (under our definition) constitute the fraction of the population that has not yet been affected by selection. We refer to the direct effect of mutation on the evolutionary dynamics when it concerns the average fitness impact that individuals experience from interactions with the mutants, i.e., the instantaneous impact of mutational variation before selection acts upon these mutants. Generally, the fitness impact of mutation goes beyond that of the mutants alone as long as they (faithfully) produce descendants. We refer to this latter impact as the indirect effect of the mutants when it concerns the average fitness impact that individuals experience from interactions with descendants of mutants.

In this paper, we show that the distinction between direct and indirect fitness effects is useful in providing insight in the impact of mutation-generated variation on the evolution of cooperation. Our results show that (1) earlier findings on the impact of behavioral variation on the evolution of cooperation are no more than special cases where there is a positive direct effect of mutation; (2) cooperation can evolve through an indirect effect of mutation-generated variation even if unconditional defectors benefit most from the direct effect; and (3) in cases with positive indirect effects of mutation, the selected strategies in the population exhibit exploitation of cooperative acts.

## 2. Model

We use the evolutionary game version of the finitely repeated Prisoner's Dilemma game (frPD; [18,19]) to study the impact of different mutation regimes. In the Prisoner's Dilemma (PD) game, individuals engage in pairwise interactions where they choose to cooperate (*C*) or to defect (*D*). Both players execute their action simultaneously. Individuals receive payoff dependent on their own choice, as well as the choice of their opponent, as given in the following payoff matrix:

|   | *C* | *D* |
|---|-----|-----|
| *C* | *R* | *S* |
| *D* | *T* | *P* |

The payoffs follow the relations $T > R > P > S$ and $2R > T + S$. The consequence is that defection generates higher individual payoffs *T* (>*R*) and *P* (>*S*), while mutual cooperation maximizes the common payoff. In the frPD game, the Prisoner's Dilemma game is repeated a fixed number (=*r*) of times. The total number of deterministic strategies adjusted to an frPD game is finite (i.e., $2^{2^r-1}$; [19]). Cressman [19] provides a general analysis of evolutionary frPD games in absence of mutation. The strategy that defects (=*d*) and the strategy that cooperates (=*c*) are the two strategies in a one-shot game (*r* = 1). It follows from the payoff matrix that for any population composition of the two strategies,

individuals with strategy *d* always generate a higher average payoff than individuals with strategy *c*. Consequently, strategy *d* evolves towards fixation in infinite populations, i.e., the state in which mutual defection is observed in all PD games. For much the same reason, in mutation-free evolutionary frPD games ($r > 1$) polymorphic populations composed of players with all deterministic strategies evolve towards states in which the players exclusively defect [18,19]. Unconditional defectors (*AllD*) obtain fitness dominance ($w_{AllD} \geq w_i$ for all strategies *i*; $w_i$ is the fitness of strategy *i*) during this process of convergence towards full defection [19].

　　In our model, we assume infinite, asexual populations composed of deterministic strategies. The code used for the strategies is described below. Strategy *i* individuals generate average payoff $\overline{p}_i = \sum_{j \in s} p_{ij} f_j$ from game interactions; $p_{ij}$ is the payoff that a strategy *i* individual generates in interactions with a strategy *j* individual, $f_j$ is the frequency of strategy *j*, and *s* represents the strategy set (i.e., the collection of the considered strategies). For generality, we assume that the average fitness of strategy *i* individuals is determined by the sum of payoff from the game and payoff from other "background" activities, i.e., $w_i = \overline{p}_i + K$, where background fitness *K* is the game-unassociated fitness component (in the simulations, we used positive integers for *K*) which in this paper is assumed identical for each individual. Fitness determines reproductive success but not survival abilities. Consequently, all players have, independent of their performance, the same expected number of pairwise interactions over their life time in the game.

## 2.1. Strategies

　　Strategies are defined by their responses to each possible sequence of actions of an opponent in the game. To take the work of [18,19] as a starting point in our analysis, we use the type of strategy sets proposed by Nachbar [18], which we here call '*TfTx*' sets. The famous 'Tit for Tat' (*TfT*; [20]) strategy, which starts with **C** and thereafter repeats the previous action of the opponent, is contained in '*TfTx*' sets. The strategy *TfTx* behaves as *TfT* in the first *x* rounds and unconditionally defects in the remaining ($r - x$) rounds. '*TfTx*' sets contain all *TfTx* strategies with $x = \{0, 1, \ldots , r - 1, r\}$. The extremes represent *AllD* ($x = 0$) and *TfT* ($x = r$). In absence of mutation, *AllD* evolves towards fixation in any polymorphism of the '*TfTx*' strategies [18]. Hence, in our analyses below, we use the fitness of *AllD* as the benchmark for measuring the relative success of other strategies when mutation-generated variation is introduced in the model.

　　McNamara et al. [3] altered the frPD game in that players can end the game after any round (whereby both players receive an identical payoff for each unplayed round). We do not implement this altered frPD game to avoid the consequent increase in the number of deterministic strategies (assuming the work of [18,19] can be extrapolated to the game of McNamara et al. [3]). Hence, we consider '*TfTx*' sets as analogous to the sets used by McNamara et al. [3]. To confirm this analogy, we repeated the simulations of McNamara et al. [3] using '*TfTx*' sets and retrieved qualitatively the same results in these simulations as McNamara et al. [3] did with their sets.

　　Besides the '*TfTx*' sets, we also investigate complete sets of all deterministic frPD strategies (the size of these sets increases with the number of rounds, *r*, that is played in the frPD game). We use the name *Xr* for the strategy sets that contain all deterministic strategies, whereby *r* represents the number of rounds of the frPD game. These strategy sets can be described with the following code. We encode individual strategies by strings of letters {*d*, *c*} which represent actions {**D**, **C**} in response to specific perceived actions of an opponent. The length of the string of letters depends on the number of rounds of the frPD game. The first letter of the code specifies the initial action (i.e., the action played in the first round of the game). The second and third letters of the code represent the responses to the initial action **D**, respectively **C**, of an opponent. If a third round is played in the game, we need to add four more positions to code for the response of a strategy to any of the four possible combinations of actions by the opponent in the first and second round. For this example of the 3-round game, we describe the four possible combinations as 'action sequences' {**DD**, **DC**, **CD**, **CC**}, where the action sequence describes the actions taken by the opponent in the order they occurred, i.e., **DC** means that

the opponent played defect in the first round and cooperate in the second round. Hence, the letters at the fourth to seventh positions in the code represent the responses to action sequences {**DD**, **DC**, **CD**, **CC**}. In a similar vein, for the 4-round game, letters at the eighth to fifteenth positions of the code represent responses to action sequences {**DDD**, **DDC**, **DCD**, **DCC**, **CDD**, **CDC**, **CCD**, **CCC**}, etc. As an example for the game with $r = 3$, the strategy *cddcdcc* cooperates in the first round (first letter of the string is *c*), unconditionally defects in the second round (second and third letters of the string are both *d*), and defects in the third round only if the opponent defected in the first round and cooperated in the second round (i.e., played action sequence **DC**) (fourth to seventh letters of the string are all *c*, except for the fifth letter which codes for action **D** when the opponent played action sequence **DC**). In notations of strategy groups, we use dots to mark code positions at which the strategies of the group can differ, i.e., in this notation a dot can be replaced with either *d* or *c*. A special example is formed by the strategies ({*X1*, *X2*, *X3*, *X4*} → {*d*, *dd.*, *dd.d . . .* , *dd.d . . . d . . . . . .* }) which signify the groups of strategies that exclusively defect in interactions among each other. We call these strategies "defectors". Populations exclusively composed of defectors are the only type of populations in which action **C** is never executed (i.e., full defection by all players).

*2.2. Mutation*

We assume that strategies are determined by genotypes. The mutation rate $\mu$ determines the fraction of offspring that do not carry the parental genotype. We refer to these offspring as mutants. We define $u_i$ as the fraction of mutants that express strategy *i*. The distributions of such fractions (*u*) are constants if the probabilities with which mutants express any of the possible strategies in the given strategy set are independent of the parental strategy. The *u* distributions are dependent variables of population compositions otherwise. We refer to the latter as variable *u* distributions.

*2.3. Evolutionary Dynamics*

We analyze evolutionary changes in this model using both discrete generations and continuous (overlapping) generations. In the discrete-generation model (*dgm*), the frequency $f_i'$ of strategy *i* in the next generation is determined as

$$f_i\prime = (1 - \mu)f_i \frac{w_i}{\overline{w}} + \mu u_i,\tag{1}$$

where $\overline{w} = \sum_i f_i w_i$ is the average fitness of the population, $\mu$ is the mutation rate, and $u_i$ is the fraction of mutants that carry strategy *i*. The frequency dynamics in the continuous-generation model (*cgm*) follows the replicator-mutator dynamics [21]:

$$\dot{f}_i = f_i((1 - \mu)w_i - \overline{w}) + \mu u_i\tag{2}$$

For ease of reference, all parameters of the model are summarized in Table 1.

We use the characteristic $w_{AllD} \geq w_i$ of the evolutionary attractor to set a benchmark for the evolutionary effect of mutation: recurrent mutation significantly affects evolutionary frPD games whenever the fitness relation $w_i > w_{AllD}$ is either persistently or periodically observed for at least one strategy *i*. Behavioral differences cause the differences in fitness between strategies *i* and *AllD*. Therefore, $w_i > w_{AllD}$ implies—as *AllD* is the strategy which always defects—that strategy *i* employs at least some cooperation. Beyond that the observation $w_i > w_{AllD}$ does not carry information about the frequency with which strategy *i* or the remainder of the population executes cooperation. However, frequencies of cooperative behaviors may positively correlate with mutation rates without challenging the fitness dominance of *AllD* [22]. For example, mutation could frequently generate the strategy unconditional cooperator (*AllC*). The expected outcome of an increase in mutation rate would then be an increased execution of cooperation but also an increased fitness of *AllD*. If significant effects ($w_i > w_{AllD}$) and increased cooperation co-occur, we take this as an indication that cooperation evolves

due to a change in the direction of selection. Please note that significant effects might emerge without the consequence that cooperation is amply executed.

**Table 1.** Overview of parameters in the model.

| | |
|---|---|
| $f_i$ | frequency of strategy $i$ |
| $p_{ij}$ | payoff of a strategy $i$ individual from interactions with a strategy $j$ individual, |
| $s$ | the strategy set (i.e., the collection of the considered strategies), |
| $\overline{p}_i = \sum\limits_{j \in s} p_{ij} f_j$ | average payoff from game interactions for a strategy $i$ individual, |
| $w_i = \overline{p}_i + K$ | average fitness of strategy $i$ individuals, |
| $K$ | game-unassociated fitness component, i.e., background fitness, |
| $\overline{w} = \sum\limits_{i} f_i w_i$ | average fitness of the population, |
| $\mu$ | mutation rate, |
| $u_i$ | fraction of mutants that carry strategy $I$, |
| $r$ | number of rounds in the frPD. |

### 2.4. Simulation Statistics

Simulations comprise the *Xr* sets {*X1, X2, X3, X4*} which contain, respectively, {2, 8, 128, 32,768} strategies. For technical reasons, due to the size of the set, only *dgm*-simulations were performed for the *X4* set.

In our simulations, we sample the following behavioral statistics. The average cooperation in round $x$ is given by $\overline{C}_x = \sum\limits_{i,j \in s} a_{x,ij} f_i f_j$, whereby $a_{x,ij} = 1$ if strategy $i$ cooperates in round $x$ against strategy $j$ and $a_{x,ij} = 0$ otherwise. The average cooperation per frPD game is given by $\overline{C} = \sum\limits_{i=1}^{r} \overline{C}_i$ (in case of non-equilibrium dynamics, $\overline{C}$ is averaged over specified ranges of generations). As $0 \le \overline{C} \le r$, a population with $\overline{C} \sim 1$ is interpreted as fairly cooperative if $r = 1$ and as fairly uncooperative if $r = 100$. For comparisons of evolutionary frPD with different $r$-values, we thus use the average number of *C* executions per Prisoner's Dilemma game, $\frac{\overline{C}}{r}$. The average payoff per frPD game generated with payoff *P* is given by $\overline{p_P} = P \sum\limits_{i,j \in s} a_{P,ij} f_i f_j$ whereby $a_{P,ij}$ is the number of times strategy $i$ generates payoff *P* from strategy $j$. For payoffs *T*, *S*, and *R*, we analogously define the averages $\overline{p_T}$, $\overline{p_S}$, and $\overline{p_R}$.

### 3. Results

In the absence of recurrent mutation, *AllD* obtains fitness dominance at the evolutionary attractors for '*TfTx*' strategy sets [18] and for *Xr* strategy sets [18,19]. Hence, in our analyses, we use the fitness of *AllD* as the benchmark for measuring the relative success of other strategies when mutation-generated variation is introduced in the model. Specifically, the evolutionary impacts of recurrent mutation are significant whenever strategies persistently or periodically exceed the fitness of *AllD* ($w_i > w_{AllD}$). In the first subsection, we analyze the contributions to the significant impacts that result from direct effects of mutation and from the indirect effects of mutation-induced population compositions. In the second subsection, simulations of *Xr*-populations are used to assess the relevance of indirect effects in absence of direct effects.

### 3.1. Effects of Mutation on the Evolution of 'TfTx' Sets and of Xr Sets

Populations can be subdivided in fractions $\mu$ of mutants and $(1 - \mu)$ of non-mutants. Strategy $i$ generates average payoff $\pi_i = \sum\limits_{j \in s} u_j p_{ij}$ from interactions with mutants. The average fitness of strategy $i$

individuals is $w_i = (1 - \mu)\theta_i + \mu\pi_i + K$, where $\theta_i$ is the average payoff generated from interactions with non-mutants. We interpret the occurrence of inequalities $\pi_i > \pi_{AllD}$ as direct effects and of inequalities $\theta_i > \theta_{AllD}$ as indirect effects if they coincide with the observation of significant effects ($w_i > w_{AllD}$). Direct effects and indirect effects are not mutually exclusive. In the following, the discussion on indirect effects focuses on their emergence for cases when direct effects are excluded (i.e., $\pi_{AllD} \geq \pi_i$ for all $i \in s$ and at all population compositions).

For constant $u$ distributions, strategies generate fixed returns $\pi_i$, i.e., the payoff from interactions with mutants is independent of the population composition. Then, direct effects can be excluded if $(\pi_i - \pi_{AllD}) \leq 0$ for all strategies $i$. Direct effects can occur (and are inevitable for sufficiently high $\mu$-values) if $(\pi_i - \pi_{AllD}) > 0$ for at least one strategy $i$. For variable $u$ distributions, the averages $\pi_i$ are functions of population compositions. These compositions are also functions of the mutation rate $\mu$. As a consequence, there is no simple expression for when direct effects can emerge. We focus on analytical results assuming constant $u$ distributions and only briefly discuss the more complicated case of variable $u$ distributions.

For '*TfTx*' sets, the difference in performance between *TfTx* and *AllD* in interactions with mutants can be given by the recursion $\pi_{TfTx} - \pi_{AllD} = \sum_{i=0}^{x-1} (\pi_{TfTi+1} - \pi_{TfTi})$ (note that in '*TfTx*' notation, *AllD* is *TfT*0). The adjacent strategies *TfTx*$-1$ and *TfTx* perform identically with mutants expressing strategies *TfTy* for which $y \leq x - 2$. *TfTx* individuals generate one additional round of mutual cooperation from interactions with mutants expressing strategies *TfTy* for which $y \geq x$. *TfTx* individuals are exploited by *TfTx*$-1$ mutants at a single occasion, and they do not exploit *TfTx* mutants in round $x + 1$. Consequently, strategy *TfTx* is more effective in interactions with mutants than *TfTx*$-1$ ($\pi_{TfTx} - \pi_{TfTx-1} > 0$) if

$$\sum_{i=x}^{r} u_{TfTi} \, (\boldsymbol{R} - \boldsymbol{P}) \; > \; u_{TfTx\,-1} \, (\boldsymbol{P} - \boldsymbol{S}) \; + \; u_{TfTx} \, (\boldsymbol{T} - \boldsymbol{R}),$$

From this inequality it follows that for uniform '*TfTx*'-$u$ distributions (i.e., $u_{TfT0} = u_{TfT1} = \ldots = u_{TfTr}$), the distribution of the $\pi$ values has a single peak at $\pi_{TfTx}$ whereby $x$ is the highest integer for which inequality $(r + 1 - x)(\boldsymbol{R} - \boldsymbol{P}) > \boldsymbol{T} - \boldsymbol{S}$ is satisfied. As a consequence, direct effects can be obtained for uniform distributions by manipulating $\mu$ if $r(\boldsymbol{R} - \boldsymbol{P}) > \boldsymbol{T} - \boldsymbol{S}$ (i.e., $\pi_{TfT1} - \pi_{AllD} > 0$). The expectation that changes in conditions yielding increased $x$-values also result in increased execution of cooperation at the evolutionary equilibrium, was confirmed in a set of simulations.

For $Xr$ sets, the following property should be noted. The response $\rho_{ij}$ is the action sequence (of length $r$) that strategy $i$ triggers from strategy $j$, and $\rho_i$ is the entire set of responses $\rho_{ij}$ ($j \in Xr$) of strategy $i$. Given the comprehensiveness of $Xr$ sets it follows that—for arbitrary set $\rho_i$ ($i \in Xr$)—the same number of respective responses is found for each of the $2^r$ action sequences (i.e., $\rho_i$ and $\rho_j$ ($i \neq j$) are two permutations of the same set of sequences). The consequence is that, with uniform $u$ distributions, the mean behaviors of mutants are not influenced by the strategy of the opponents. In that case, it can be inferred from the payoff dominance of $\boldsymbol{D}$ over $\boldsymbol{C}$ that *AllD* generates the absolute highest mean payoff from mutants ($\pi_{AllD} > \pi_i$).

For the uniform distributions analyzed above, mean behaviors of mutants are not influenced by the strategy of the opponents. We refer to such $u$ distributions with unbiased average mutant behaviors as symmetric and to alternative $u$ distributions with biased average mutant behaviors as asymmetric. This distinction is useful because not only uniform $u$ distributions of $Xr$ sets are symmetric. For example, any distribution with uniform $u_i$ values for the conditional strategies is symmetric because the behavior of unconditional strategies is not influenced by the opponent. In the Appendix A, we define the space of symmetric $u$ distributions. Note, for both symmetric and asymmetric distributions, increasing the share of unconditional strategies tends to favor $\pi_{AllD}$ as *AllD* expresses best response behavior to unconditional strategies. As outlined for the uniform distributions, direct effects can be excluded for all symmetric distributions. Hence, direct effects emerge only if strategies can trigger distinct mean mutant behaviors (i.e., the key characteristic of asymmetric distributions).

Symmetric *u* distributions for *Xr* sets are a special case. The '*TfTx*' sets—as shown above—allow for direct effects, and they represent asymmetric distributions (the $u_i$ values of '*TfTx*' sets are formed from *u* distributions of *Xr* sets by setting the $u_i$ values to zero for strategies outside the '*TfTx*' sets). It is apparent, for direct effects to occur, that average encounters with mutants should be inefficient for *AllD* but efficient for certain other strategies, i.e., mutants should tend to conditionally defect in interactions with *AllD* and should tend to conditionally cooperate in interactions with certain other strategies. Examples are distributions (such as '*TfTx*') for which mutants tend to express reciprocal behaviors [20,23].

### 3.2. Simulations of Xr-Populations

To gain insight in indirect effects, we performed simulations using the *Xr* strategy sets {*X1*, *X2*, *X3*, *X4*} with uniform *u* distributions. As discussed in the previous subsection, direct effects are excluded with uniform *u* distributions. A set of simulations was performed with fixed parameters $K = 0$ and {*T*, *S*, *R*} = {5, 0, 3}, while varying mutual defection payoff $P = \{0.05, 0.3, 1\}$ and mutation rates $\mu = \{0.0001, 0.001, 0.01, 0.1\}$. For these parameter combinations, Table 2 shows whether populations evolve to an equilibrium or not (equilibrium conditions are described in the Appendix B). For all settings, {*X1*, *X2*}-populations (i.e., playing the one-round and the two-round game) evolve to equilibrium (Table 2). The table shows that for $P = \{0.05, 0.3\}$, no equilibrium is attained in the evolution of certain *X3*-populations and of certain *X4*-populations.

**Table 2.** Observed type of dynamics in the final phases of simulations of four *Xr* sets (top row) at three mutual defection payoffs (first column). For each {*Xr*, *P*} combination, simulations were performed at the mutation rates $\mu = \{0.0001, 0.001, 0.01, 0.1\}$. In this alignment, letters {*n*, *e*, *E*} of the four-digit strings represents the dynamics found in simulations at respective rate; *n*: non-equilibrium dynamics, *e*: equilibrium in which only defectors obtain above average fitness, and *E*: equilibrium in which non-defectors obtain above average fitness (for example, *eeeE* means that equilibrium is found at all four rates whereby non-defectors attain above average fitness only at rate $\mu = 0.1$). Fixed parameters: $T = 5$, $S = 0$, $R = 3$, $K = 0$.

|             | *X1* | *X2* | *X3* | *X4* |
|-------------|------|------|------|------|
| $P = 0.05$  | *eeee* | *eeee* | *nnEE* | *nnEE* |
| $P = 0.3$   | *eeee* | *eeee* | *nEEE* | *nnnE* |
| $P = 1$     | *eeee* | *eeee* | *eeeE* | *eeeE* |

The equilibrium populations described in Table 2 are dominated by *AllD* (i.e., $f_{AllD} > f_i$ for $i \neq AllD$)—this characteristic applies to all observed equilibrium populations in our study. Furthermore, all observed equilibrium strategy frequencies $f_i$ are identical for both continuous and discrete-generation models. At equilibrium, dominance of *AllD* implies that the strategy also has fitness dominance. We do not find persistent indirect effects in the populations that do not reach equilibrium. Consequently, we do not find persistent indirect effects in the simulations.

For $P = 1$, the {*X1*, *X2*, *X3*, *X4*}-populations evolve to equilibrium for all mutation rates (Table 2). For rates $\mu = \{0.001, 0.01, 0.1\}$, Table 3a shows the average number of *C* executions per Prisoner's Dilemma game ($\frac{\overline{C}}{r}$) in these equilibrium populations. For each setting, these averages increase with mutation rates. The $\frac{\overline{C}}{r}$-values of *X1*-populations (Table 3a) are only slightly higher than the inflow of cooperator (*c*) mutants (~$0.5\mu$). The execution of cooperation can thus be attributed to *c*-mutants. The table shows for each mutation rate that $\frac{\overline{C}}{r}$-values of {*X2*, *X3*, *X4*}-populations are approximately three times higher than those of *X1*. We attribute this difference to the fact that sets {*X2*, *X3*, *X4*} contain conditional strategies. Table 3b shows that evolution to an equilibrium is found in simulations of *X3*-populations using the two background fitness values $K = \{0, 5, 20\}$. Along $K = \{0, 5, 20\}$ we find an increase in mean cooperation for each rate $\mu$ (Table 3b).

**Table 3.** Average amount of executed *C* actions per PD game ($\frac{\overline{C}}{r}$) found in populations at equilibrium, as a function of mutation rate. The top row gives mutation rates, first column the *Xr* sets, and second column the background fitness values *K*. Part a: varying strategy set for *K* = 0; part b: varying background fitness for strategy set *X3*. The $\frac{\overline{C}}{r}$-values from equilibrium populations in which non-defectors obtain above average fitness are given in italic. Fixed parameters: ***P*** = 1, ***T*** = 5, ***S*** = 0, ***R*** = 3.

|  |  |  | $\mu$ = 0.001 | $\mu$ = 0.01 | $\mu$ = 0.1 |
|---|---|---|---|---|---|
| a |  |  |  |  |  |
|  | *X1* | *K* = 0 | 0.0005 | 0.0051 | 0.0577 |
|  | *X2* | *K* = 0 | 0.012 | 0.012 | 0.141 |
|  | *X3* | *K* = 0 | 0.0014 | 0.0148 | *0.1844* |
|  | *X4* | *K* = 0 | 0.0013 | 0.013 | *0.1712* |
| b |  |  |  |  |  |
|  | *X3* | *K* = 0 | 0.0014 | 0.0148 | *0.1844* |
|  | *X3* | *K* = 5 | 0.0038 | 0.0382 | *0.3294* |
|  | *X3* | *K* = 20 | 0.0109 | 0.1002 | *0.43* |

The observed cooperation in the populations of Table 3 is maintained by mutation-selection balance because direct and indirect effects are absent. This interpretation of the $\frac{\overline{C}}{r}$-data is straightforward for the *X1*-populations. For the populations with repeated games, cooperation can be argued to be disadvantageous because 'non-*AllD*'-individuals would increase their fitness by substituting their strategy for *AllD*. However, we emphasize that in several {*X3*, *X4*}-populations, non-defectors obtain above-average fitness at equilibrium (Tables 2 and 3). The potential for the evolution of conditional behavior in repeated games ({*X2*, *X3*, *X4*}) seems to reduce selection against cooperation (as cooperation levels are higher for these sets than in *X1*; see Table 3a). As expected, a similar effect can be attributed to increasing background fitness *K* (Table 3b).

Table 2 shows that for the lowest mutual defection payoff (***P*** = 0.05), {*X3*, *X4*}-populations do not converge to equilibrium in the simulations with the two lowest mutation rates. For the intermediate ***P***-value of Table 2, this phenomenon is also observed for *X3*-populations at the lowest rate and for *X4*-populations at the three lowest rates. With its 256 times smaller set size, the *X3*-populations are more convenient to study. This is why we mainly study non-equilibrium behavior in *X3*-populations.

For ***P*** = 0.05, Figure 1a shows the mean execution of cooperation per frPD game ($\overline{C}$) along $\mu$ = {0.00001, 0.0001, 0.001, 0.01, 0.1}. For the lowest rate and for the two highest rates, these means are sampled at equilibrium. As mentioned, the equilibrium frequencies are not affected by the choice of the generation model (i.e., *dgm* or *cgm*). Hence, the $\overline{C}$-values are identical in Figure 1a for each of these rates. After transient phases, the populations at rates $\mu$ = {0.0001, 0.001} evolve in cycles. As an example, consider the strategy dynamics at rate $\mu$ = 0.001 in Figure 1b for *dgm* and in Figure 1c for *cgm*. Table 4 lists the strategies with max($f_i$) > 0.1 during the cycles for these two figures. For mutation rates $\mu$ = {0.0001, 0.001}, the $\overline{C}$-values in Figure 1a are averaged over one cycle period. The $\overline{C}$-values are identical if populations are initialized with $f_{AllD}$ = 1 and with a uniform frequency distribution. For both mutation rates, the averages $\overline{C}$ are higher if sampled over *dgm*-cycles than if sampled over *cgm*-cycles (Figure 1a). The figure also shows that for both models, the $\overline{C}$-values are higher in the cycling populations than for the equilibrium populations at $\mu = 10^{-5}$. The $\overline{C}$-values are higher than the equilibrium-values found at the higher rate $\mu$ = 0.01 for the *dgm* at rates $\mu$ = {0.0001, 0.001} and for the *cgm* at rate $\mu$ = 0.001 (Figure 1a). Consequently, for both types of generation models, an optimum in $\overline{C}$ exists within the interval $10^{-5} < \mu < 0.01$.

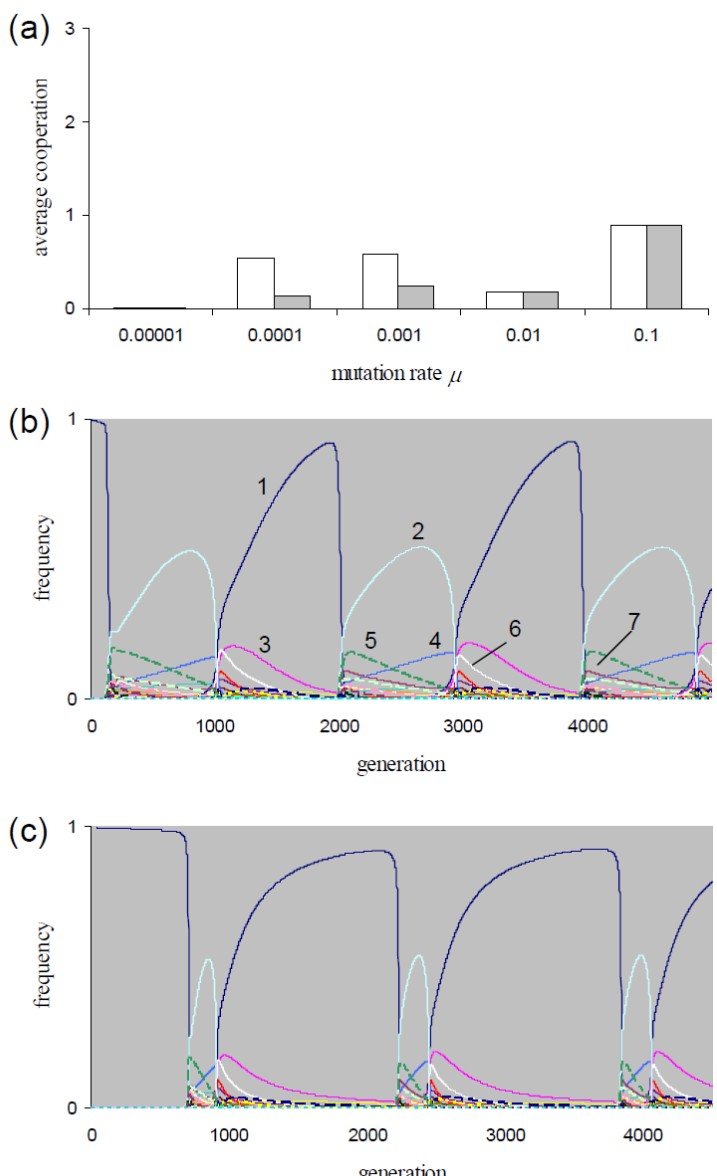

**Figure 1.** (**a**) Average execution of cooperation per game ($\overline{C}$) as function of mutation rate in *X3*-simulations of *dgm*-populations (white) and of *cgm*-populations (gray). For rates $\mu = \{10^{-5}, 0.01, 0.1\}$, averages are sampled at equilibrium, and for rates $\mu = \{0.0001, 0.001\}$, averages are sampled over a cycle period (see panel (**b,c**)). Fixed parameters: $P = 0.05$, $T = 5$, $S = 0$, $R = 3$, $K = 0$; (**b**) For the conditions of panel (**a**), the frequency dynamics of an evolving *dgm*-population at rate $\mu = 0.001$. 1: *ddddddd* (*AllD*); 2: *cdddddd* (*TfT*1); 3: *dddddddc*; 4: *ddcdddd*; 5: *cdcdddd* (*TfT*2); 6: *dcddddd*; 7: *dccdddd*; (**c**) For the conditions of panel (**a**), the frequency dynamics of an evolving *cgm*-population at rate $\mu = 0.001$. Panel (**b,c**) use the same line code.

For the two mutation rates $\mu = \{0.0001, 0.001\}$, we tested the sensitivity of the cycling dynamics in *dgm*-populations to the choice of background fitness $K$. The populations show cycling dynamics if $K \leq \{45, 3\}$ ($\rightarrow \mu = \{0.0001, 0.001\}$) and evolve to equilibrium for higher $K$-values. The *X3*-populations showing non-equilibrium dynamics in our simulations evolve in cycles (and show periodic indirect effects).

The strategy dynamics in Figure 1b,c resemble those in the corresponding simulations with the lower mutation rate $\mu = 0.0001$. All four cycles show (as in Figure 1b,c) alterations of phases with dominance of *AllD* followed by phases with dominance of *TfT*1 (*cddddddd*). As can be inferred from these dynamics, *AllD* respectively *TfT*1 have the highest fitness when invading the populations.

Consequently, these populations express periodic indirect effects. In Figure 2, we give behavioral statistics from the simulation of Figure 1b. Figure 2a shows the dynamics of the mean number of executed $C$ actions for each round of the game ($\overline{C}_i$, $i = \{1, 2, 3\}$). Cooperation is more intensively executed during *TfT*1 phases, especially in round 1 (Figure 2a). The relatively longer *TfT*1 phase durations in the *dgm*-populations (compare Figure 1b with Figure 1c) explain that $\overline{C}$-values are higher in *dgm*-populations than in corresponding *cgm*-populations (Figure 1a at $\mu = \{0.0001, 0.001\}$).

**Table 4.** List of strategies that obtain peak frequencies higher than 0.1 ($\max(f_i) > 0.1$) within the cycle phases of the *dgm*-dynamics depicted in Figure 1b. Code representation (conventional name in brackets) of the strategies is given in the second column. The peak frequency within the cycle phases is given in the third column. The peak frequency within the cycle phases of the *cgm*-dynamics of Figure 1c is given in the fourth column.

|   | Strategy | $\max(f_i)$ | $\max(f_i)$ |
|---|----------|-------------|-------------|
| 1 | *ddddddd* (*AllD*) | 0.921 | 0.921 |
| 2 | *cdddddd* (*TfT*1) | 0.544 | 0.544 |
| 3 | *dddddc* | 0.201 | 0.201 |
| 4 | *ddcdddd* | 0.168 | 0.168 |
| 5 | *cdcdddd* (*TfT*2) | 0.166 | 0.165 |
| 6 | *dcddddd* | 0.159 | 0.157 |
| 7 | *dccdddd* | 0.103 | 0.102 |

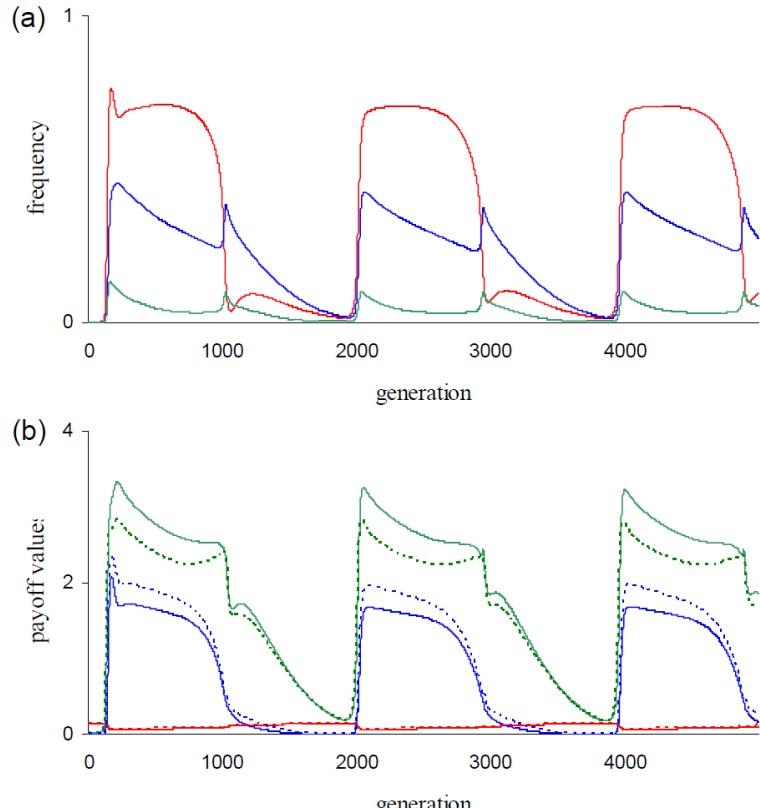

**Figure 2.** Both panels show behavioral statistics of the evolving populations in Figure 1b. (**a**) Average execution of cooperation in round 1 ($\overline{C}_1$: red), in round 2 ($\overline{C}_2$ : blue), and in round 3 ($\overline{C}_3$ : green) as function of time; (**b**) Average payoff generated per game with payoff $P$ (red), payoff $T$ (green), and payoff $R$ (blue) as function of time. Dashed lines represent respective expected values (i.e., $P \sum\limits_{i \in \{1,2,3\}} (1 - \overline{C}_i)^2$, $T \sum\limits_{i \in \{1,2,3\}} (1 - \overline{C}_i)\overline{C}_i$, and $R \sum\limits_{i \in \{1,2,3\}} \overline{C}_i^2$) and solid lines represent respective observed values (i.e., $\overline{p}_P$, $\overline{p}_T$, and $\overline{p}_R$).

For the three payoffs *P*, *T*, and *R*, Figure 2b shows the dynamics of the mean payoff values per frPD game (i.e., $\overline{p_P}$, $\overline{p_T}$, and $\overline{p_R}$). Steep increases in the generation of *T*-payoffs (Figure 2b) mark the onset of invasions by *TfT*1 (Figure 1b). Defectors like *AllD* generate this payoff in the first round when interacting with *TfT*1 and defectors are the dominant opponents of this strategy at the onset of invasions (Figure 1b). The increase in the generation of *T*-payoffs is therefore partly explained by defectors triggering this payoff from *TfT*1. For *TfT*1, these first round interactions seem disadvantageous, but this disadvantage is evidently compensated because *TfT*1 invades.

Figure 2b additionally shows the dynamics of expected average payoff values generated per game from receiving payoff *P*, *T*, or *R* (i.e., $P \sum\limits_{i \in \{1,2,3\}} (1 - \overline{C_i})^2$, $T \sum\limits_{i \in \{1,2,3\}} (1 - \overline{C_i})\overline{C_i}$, and $R \sum\limits_{i \in \{1,2,3\}} \overline{C_i}^2$). For payoff *T*, the observed value is higher than the expected value (Figure 2b) over the dominance phase of *TfT*1 (Figure 1b). These differences between observed and expected values are caused by the conditional behaviors in rounds 2 and 3. Hence, we propose that the invasions of *TfT*1 are fueled by triggering *T*-payoffs in these rounds. At the onset of invasions, *AllD* is the dominant strategy (Figure 1b) and defection is the predominant behavior (Figure 2). Defectors (in contrast to non-defectors) are not penalized when interacting with *AllD* and they can therefore be expected to perform better than other strategies in *AllD*-dominated populations. The strategy *TfT*1 generates *T*-payoffs from the twelve defectors {*ddcd . . .* , *dddd.c.*}. Game interactions between these defectors and *TfT*1 indeed significantly contribute (data not shown) to the increases of $\overline{p_T}$ (Figure 2b).

In the appendix, we derive the invasion condition for a single *TfT*1-individual in a population state with full defection. We find that such invasion occurs if the combined frequency of defectors {*ddcd . . .* , *dddd.c.*} exceeds $(P − S)/(T − P)$ (~0.01 in the simulation of Figure 1b). This condition is fulfilled over the entire cycle period in Figure 1b, but the population state deviates from full defection due to mutation. In this state, *AllD* obtains the highest benefit from interactions with mutants (i.e., $\mu(\pi_{AllD} − \pi_{TfT1}) > 0$). Thus, the invasion conditions in the simulations should be more stringent than those derived in the appendix. Before the onset of the invasions, the population does converge towards a state of full defection (Figure 2) and thus towards the conditions underlying the analysis in the appendix. In our opinion, the invasions of *TfT*1 in the simulations are fueled by interactions with defectors {*ddcd . . .* , *dddd.c.*}, just like in the analysis. That *AllD* subsequently regains dominance, thereby closing the cycle (Figure 1b,c), is in line with the expectations from the selection dynamics of evolutionary frPD games [18,19].

In Tables 2 and 3, we mark the equilibria (*E* in Table 2 and italic numbers in Table 3) in which non-defectors have above-average fitness. The strategy *TfT*1 has the highest fitness among the non-defectors in these equilibria. Furthermore, these equilibria emerge at the higher mutation rates (Tables 2 and 3) possibly because mutation benefits *TfT*1 (e.g., by generating defectors {*ddcd . . .* , *dddd.c.*} opponents) in these equilibria. However, invasion by this strategy is prevented also because *AllD* is the strategy that benefits most from interactions with mutants ($\mu(\pi_{AllD} − \pi_{TfT1}) > 0$).

The *X4*-simulations are more computation-intensive than the *X3*-simulations, and we restricted these simulations to $10^4$ generations due to constraints on computation time. Consequently, the data obtained do not allow definitive conclusions on the nature of non-equilibrium *X4*-dynamics. Over the simulation periods, chaotic dynamics occurs for the *X4*-populations with non-equilibrium dynamics in Table 2. For example, in Figure 3, the *X4*-frequency dynamics at {*P*, $\mu$} = {0.3, 0.01} exhibits a transient period of ~2000 generations, after which alternations of dominance by strategies {*AllD*, *ddcddddddddddd*, *TfT*1, *dddddcddddddddd*} emerge. The population therefore expresses periodic indirect effects. As in Figure 1b,c, strategies *AllD* and *TfT*1 in Figure 3 become periodically dominant, with dominant phases of strategies {*AllD*, *TfT*1, *dddddcddddddddd*} that have fairly regular phase lengths (Figure 3).

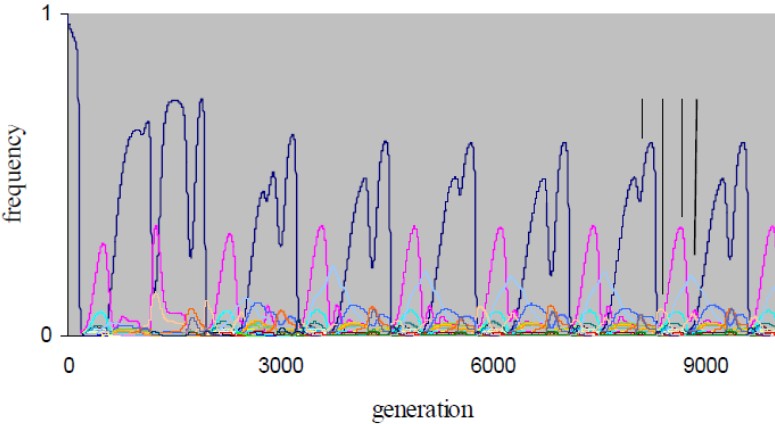

**Figure 3.** The frequency dynamics of an *X4*-simulation at {*P*, *μ*} = {0.3, 0.01}; other conditions as in Figure 1. The four vertical lines indicate the position of a respective dominance phase of strategies {*AllD*, *ddcddddddddddddd*, *TfT*1, *dddddcdddddddddd*}.

For {*X1*, *X2*, *X3*, *X4*}, Table 2 shows that all populations evolve to equilibrium in the two smallest sets {*X1*, *X2*}, and non-equilibrium dynamics occurs more frequently when going from set *X3* to set *X4* (e.g., the X3-population evolves to equilibrium under the conditions of Figure 3). We interpret this observation as an indication that increasing the number of rounds (*r*) increases the parameter range for which periodic indirect effects emerge. This interpretation meets our intuition because strategy *TfT*1 generates one *T*-payoff from X2-defector *ddc*, two *T*-payoffs from X3-defectors *ddcdc..*, and three *T*-payoffs from X4-defectors *ddcdc..d . . . c . . . .*

## 4. Discussion

Whereas empirical ecologists typically observe wide behavioral variation, theoretical ecologists tend to ignore or minimize behavioral variation in their models in order to make their analyses tractable. In this paper, we provide a method to analyze effects of behavioral variation on evolutionary dynamics, and apply it to the evolution of cooperation. We present a model in which behavioral variation is on the one hand subject to a restriction because probabilistic strategies are excluded, yet on the other hand comprehensive because all deterministic strategies are taken into account (see [24–27] for earlier studies of evolutionary repeated games with comprehensive strategy sets). We first discuss the method of analysis, and then when and how behavioral variation affects the evolution of cooperation.

### 4.1. Direct and Indirect Effects of Mutation-Generated Variation

We consider direct and indirect effects of mutation in frequency dependent selection environment. Direct effects are fitness effects that emerge from (game) interactions with mutants and indirect effects are fitness effects that emerge from interactions with descendants of mutants. Direct effects are mainly determined by the mutation regime (as reflected in the *u* distribution of mutants over all possible strategies). Indirect effects are additionally influenced by selection on the progeny of mutants. Both in direct and indirect effects invading strategies depend on the presence of other strategies (see [27,28] for analysis of invasions that depend on other strategies).

In our analysis of direct and indirect effects of mutation in asexual populations, we take advantage of simplifying cladistics: each mutant is the founder of a clade whereby the clade constitutes the clonal descendants that faithfully inherit the genome of the mutant. Direct effects are caused by interactions with founders and indirect effects are caused by interactions with descendants. If we had modeled sexual reproduction, then this would complicate the cladistics, as clades cross over, thereby generating new behavioral variants in another way than by mutation. Cultural transmission, however, could have similarly simple cladistics to our asexual model; a system with innovators and imitators could, to some degree, be analogous to our system with mutants and descendants (non-mutants).

To analyze direct effects for large strategy sets, we assume constant *u* distributions resulting in constant returns ($\pi_i$) from the interactions with mutants. In nature, *u* distributions are probably variable; for example, the *u* distribution is variable if mutation swaps single code positions rather than modifies entire codes/strategies (as in our study). Variable *u* distributions would complicate the analysis of direct effects because the population composition has to be considered (whereby genotype x mutation interactions, i.e., genotypes differ in their propensity to mutate, would further complicate this analysis).

We also assume that fitness differences concern differences in fertility. If we had considered fitness differences in viability, this would have complicated the determination of the $\pi_i$-values. For example, it is not clear to us whether $\pi_i$-values are still constants with constant *u* distributions. The fraction of mutants would definitely deviate from $\mu$. However, if we had considered differences in viability (rather than fertility), then we cannot think of a reason our qualitative findings with respect to the '*TfTx*' sets, the *Xr* sets, and the symmetric *u* distributions would have changed.

Indirect effects require sufficiently strong frequency-dependent selection; they only emerge when the strength of selection is high. Indeed, indirect effects do not emerge in our simulations above certain values of background fitness *K*. To demonstrate indirect effects, our method is best applied to a system with a unique attractor in the selection environment in absence of mutation. This attractor in turn is best represented by a strategy that benefits most from interactions with the mutants, because, otherwise, direct effects could blur indirect effects.

## 4.2. Direct Effects as a Mechanism Promoting the Evolution of Cooperation

We use an evolutionary game version of the finitely repeated Prisoner's Dilemma (frPD), because it has a unique attractor in absence of mutation: unconditional defector (*AllD*) [19]. This provides a straightforward criterion to test what happens when mutation is included; mutation has significant effects whenever at least one strategy persistently or periodically achieves a higher fitness than that of *AllD*.

We show conditions for direct effects when mutation produces '*TfTx*' strategies [18], which permit the evolution of cooperation. We further show that the evolution of cooperation through direct effects are excluded for *Xr* strategy sets [19] with a uniform *u* distribution. For the latter analysis, we define a class of *u* distributions, the symmetric *u* distributions (see Appendix A: uniform *u* distributions of *Xr* sets are examples of symmetric *u* distributions), for which the average (conditional) behavior of mutants is independent of the opponent strategy. Whenever the opponent strategy does not influence the average behavior of mutants, then *AllD* is the opponent strategy that receives the highest payoff from the interactions with mutants. Hence, direct effects are excluded for symmetric *u* distributions.

The '*TfTx*' sets exemplify how to create mutation regimes that can elicit direct effects resulting in the evolution of cooperation. The *TfTx* strategies are contained in the corresponding *Xr* set and the *Xr* sets are subsets of the (infinite) space of probabilistic strategies adjusted to frPD games. We introduce the symmetric *u* distributions because they constitute a boundary in the *u* distribution space: they separate the subspace where mutants cooperate most often with *AllD* from the subspace where mutants cooperate most often with another strategy. Direct effects can only emerge in the latter subspace, depending on the conditions of the frPD game and the mutation rate. Please note that the emergence of direct effects does not imply that the population evolves to a state where cooperation is amply executed (e.g., if the strategy benefiting most from the mutants is a defector).

Direct effects that result in the evolution of cooperation emerge only in a fraction of the *u* distribution space. We can only speculate about the size of this fraction. Even if this fraction is tiny, it may be important for the evolution of cooperation if *u* distributions in natural systems would fall into this category. Nevertheless, the empirical evidence for direct reciprocity (in general) is scarce [29], let alone evidence for mutation regimes. In any case, for evolutionary games with a clear attractor strategy like *AllD* for evolutionary frPD games, we conjecture that this strategy is most likely to benefit

most from the mutants. Consequently, the evolution of cooperation by direct effects is possible but we predict it is not very likely in general.

The study of McNamara et al. [3] inspired our definition of direct effects. In our view, the evolution of cooperation in the studies of [4–6] is explained by direct effects. The strategy sets used in these studies are only part of a much larger set of (deterministic) strategies, just like the '*TfTx*' sets in relation to the *Xr* sets. Furthermore, if their mutation regimes were replaced by regimes comprising broader strategy sets, then we would expect that the unconditional defectors in their evolutionary games benefit most from interactions with mutants. Hence, we think that the mutation-induced promotion of the evolution of cooperation, as described by [3–6], is a rather special outcome. If, however, behavioral variation is not caused by mutation,but culturally inherited, then cooperation may evolve under a wider set of conditions if it is true that humans choose from '*TfTx*' strategy sets and disregard most *Xr* strategies.

### 4.3. Indirect Effects as a Mechanism Promoting the Evolution of Cooperation

In simulations of *Xr* sets with uniform *u* distributions (i.e., a condition without direct effects), we observe periodic indirect effects (Figure 1b,c and Figure 3). We find several conditions where such effects emerge (Table 2) at intermediate mutation rates (Figure 1a). The periodic indirect effects all show a similar pattern (Figures 1b,c and 3): a population dominated by *AllD* is invaded by strategy *TfT*1 and vice versa, giving rise to cycles of alternating dominance. We suggest that the invasion of *TfT*1 is due to a group of defectors defined by {*ddcd . . . , dddd.c.*} (see also Appendix C). The behavior of these defectors towards other defectors is similar to the dominant behavior in *AllD*-dominated populations: play defect in all rounds of the game. Therefore, these defectors are less vulnerable to the (*AllD*-influenced) selection in such populations and decrease less (due to mutation-selection balance) than other *Xr* strategies. *TfT*1 exploits the behavioral deviations that certain defectors have from the behavior of *AllD* (see Appendix C). As a consequence, the execution of cooperation increases during *TfT*1-invasions (Figure 2a) and the average execution of cooperation can exceed the execution value expected from a mutation-selection balance (Figure 1a).

We only found periodic indirect effects. We suspect that—for evolutionary frPD games (without direct effects)—persistent indirect effects emerge only for special parameter regions, if they exist at all. This is because they require a strong enough effect of the *AllD* individuals on the fitness of others already before this strategy fully achieves fitness dominance.

The periodic indirect effects observed in our study resulted in periodic increases in the execution of the action 'cooperation' (Figure 2a). However, one may ask whether this increase constitutes co-operation in the sense of individuals mutually helping each other. If *TfT* invades a population otherwise composed by *AllD* then after the first round the *TfT*-players cooperate only with other *TfT*-players. Cooperation therefore mostly takes place among individuals with the same phenotype (i.e., *TfT* players) and the mutual cooperation payoff is generated more often than expected. Such positive assortment is known as a fundamental principle for the evolution of cooperation [30]. In our simulations, we find the opposite: decisive executions of cooperation take place between individuals with different phenotypes, i.e., *TfT*1 and defectors {*ddcd . . . , dddd.c.*}, and the mutual cooperation payoff is generated less often than expected (Figure 2b). Furthermore, in the interactions between *TfT*1 and these defectors it is not beneficial for the defectors to stick to their strategy, as they would fare better by playing unconditional defection. On the other hand, it is typical for the evolution of cooperation that the average payoff increases, as observed during *TfT*1 invasions (Figure 2b).

Strategy *TfT* invades in the large set of strategies that invade in the wake of *TfT*1 invasions (Figure 1b,c and Figure 3). This conditional cooperator might play a more pronounced role in frPD games with more than four rounds (as increasing *r* promotes the performance of *TfT*). Unfortunately, we cannot check this prediction because the sheer size of the corresponding *Xr* sets (e.g., 2,147,483,648 strategies in a game with five rounds) makes running simulations with five or more rounds unfeasible.

## 5. Conclusions

By analyzing direct effects, we explain the phenomenon of mutation-promoted evolution of cooperation, described previously by [3–6]. However, we argue that this phenomenon is probably a rather special case; the strategy favored by selection without mutation—*AllD* in our study—is most likely also the strategy with the highest benefit from interactions with mutants (direct effect). In such cases, cooperation can still evolve as an indirect effect of mutation-generated variation under a limited set of conditions. The resulting cooperation dynamics, however, shows exploitation of cooperative acts rather than mutual cooperation. Indeed, evolution of cooperation in PD games typically requires a mechanism that induces assortativity in couples of players with cooperative or defective strategies [31], in contrast to models of the evolution of collaboration (e.g., [32,33]).

The study of McNamara et al. [3] is seminal in highlighting the importance of behavioral variation in evolutionary dynamics [2]. Theoreticians tend to avoid this topic because behavioral variation complicates model analysis. To facilitate such analysis, our method to separate direct and indirect effects of behavioral variation is a useful approach to assess if and when behavioral variation is important in evolutionary dynamics.

**Author Contributions:** M.S. and M.E. designed the study, M.S. carried out the simulations and analytical work and wrote the first version of the manuscript, M.E. supervised and gave feedback and wrote the final version of the manuscript.

**Funding:** This paper is part of the research project 'Simultaneous evolution of social norms and Social Behaviour: A combined theoretical and experimental approach' funded by The Netherlands Organisation for Scientific Research (NWO).

**Acknowledgments:** We thank Maurice W. Sabelis (deceased 7 January 2015) for his guidance and supervision during this project.

**Conflicts of Interest:** The authors declare no conflicts of interest.

## Appendix A. The Symmetric u Distributions

A *u* distribution is symmetric if the average conditional game behavior of mutants is independent of the game behavior that the opponent expresses. That is, given a mutant sampled from this distribution, if this mutant expresses conditional behavior in arbitrary round *x* then the probability that this mutant defects in round *x* is independent of the action sequence the opponent played in the previous rounds. In the following, we demonstrate the symmetric *u* distributions.

Game behavior can be conditional from round 2 onwards. Strategies carrying *dd* or *cc* at code positions 2 and 3 express unconditional behavior in round 2. Strategies carrying *dc* or *cd* express conditional behavior in this round. Similarly, strategies carrying *dddd* or *cccc* at positions 4 to 7 express unconditional behavior in round 3. The strategies with a different code at these positions express conditional behavior in round 3. Also, for the higher rounds, two codes determine unconditional behavior and the remaining codes determine unconditional behavior in that round.

Constant *u* distributions are symmetric if each round behavior mutates according to the following rule. In round *x*, the mutant expresses unconditional defection with probability $u_{x1}$, unconditional cooperation with probability $u_{x2}$, and conditional behavior with probability $u_{x3}$ (=$1 - u_{x1} - u_{x2}$). In case of the latter mutation event, the probability that the mutant carries either code determining conditional round *x* behavior requires the following property: the chance that the mutant defects (cooperates) in this round is independent of the previous action sequence played by the opponent. This condition is given if either of these codes is carried by the mutant with equal probability.

Variable *u* distributions are symmetric if each round behavior mutates according to the following rule. Mutation treats parental genotypes with unconditional round *x* defection, with unconditional round *x* cooperation, and with conditional round *x* behavior differently. Unconditional round *x* defection (cooperation) is faithfully inherited with probability $1 - u_{xD}$ ($1 - u_{xC}$) and mutates otherwise. Conditional round *x* behavior always mutates. Dependent on the parental genotype, we therefore find three forms of mutation events. For each form, mutation proceeds analogously as described

for constant symmetric *u* distributions (whereby the three probabilities $\{u_{x1}, u_{x2}, u_{x3}\}$ can be distinct between the forms).

**Appendix B. Equilibrium Conditions**

We assess equilibrium in {*X1*, *X2*, *X3*}-populations as reached if all differences $(f_i{}' - f_i)$ in the *dgm* are less than $10^{-8}$ between consecutive generations. Note, the corresponding *cgm*-populations evolve to the same frequency distributions for all observed equilibria. The approached equilibria are identical for initial *X1*-populations $\{f_D = 0, f_D = 0.5, f_D = 1\}$. The equilibrium {*X2*, *X3*}-populations are identical if initiated with $f_{AllD} = 1$ or if initiated with uniform frequency distributions. Only the *dgm*-model is implemented to simulate *X4*-populations and all *X4*-populations are initiated with $f_{AllD} = 1$. *X4*-equilibrium is assumed if all differences $f_i{}' - f_i$ decrease over consecutive generations for a period of 2000 generations.

**Appendix C. Invasion Condition of TfT1 in X3-Defector Populations**

We study the fitness of a *TfT*1 individual in populations otherwise composed of defectors (full defection). Defectors B are the strategies {*ddcd* . . . , *dddd.c*.}. Defectors A are the remaining defectors of the *X3* strategy set. Strategy *TfT*1 earns payoff $(S + 2P)$ from interactions with defector A individuals, payoff $(S + 2T)$ from interactions with defector B subgroup *ddcd.c.*, and payoff $(S + P + T)$ from interactions with the other defector B subgroup. The combined frequency of defectors B strategies is defined as $f_B$. We calculate the most stringent condition for invasion of *TfT*1, i.e., assuming that the defectors *ddcd.c.*, from which *TfT*1 generates the higher payoff, are absent. Then, *TfT*1 individuals should obtain above average payoffs if $f_B (S + P + T) + (1 - f_B) (S + 2P) > 3P$. Consequently, negligibly small $f_{TfT1}$-values increase if $f_B > (P - S)/(T - P)$.

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
