# Peer review of "When and How Does Mutation-Generated Variation Promote the Evolution of Cooperation?"

_games, doi:10.3390/g10010004_

Reviewer 1 Report

Tha paper under review tries to understand the role of mutations in the emergence and evolution of cooperation. In order to do that, the authors consider an asexual population playing a Prisoner's Dilemma Game with deterministic strategies and mutations.

The idea is interesting, the study and analyses appear to have been accomplished properly, but I have to admit I have found very difficult understanding the description of the model, and how it works.

In particular, the authors should clarify much bettter the meaning of the strategies as defined in Subsec. 2.3 (page 4 of 16). Why the strategy (for r=3) cddcdcc means cooperation in the first round, unconditional defection at the second round, and defection at the third round only if the opponent defected in the first and cooperated in the second round? With such notation, how can one distinguish the unconditional moves from the ones depending on the opponent's choises, and where are the opponent's choises indicated? The definition of the TfTx strategies is clearer, but in this case it would be good to explain why adding this peculiar type of strategies: why does utilizing that help us to understand better the behaviour of the model?

Moreover, I would suggest to anticipate this Subsection (2.3 Strategies) at the end of the introductory paragraph of section 2, before the Subsec. 2.1 (Analysis), in order that the reader have a clear idea of the structure of the game when introduced to the analysis and statistics.

More in general, there are many parameters and quantities which define the model (p, u, f, w...), so it is easy for a non-expert reader to get lost. Finally, why adding the quantity K? Is this background fitness really necessary to the model? If yes, please justify more deeply its introduction. 

The rest of the paper would result more understandable and easy to read once the definition of the model has been rewritten more clearly.

My last concern is the definition of the direct-undirect effects, being the first ones related to the interactions with the "founders" (the first to have a certain mutation, if I have understood correctly), and the second ones with its descendants. Hpw is it related to direct-undirect reciprocity (I reciprocate the action the opponent adopts with me withhim/herself, or with other agents)? This also need clarifications.

I repute that this paper can be taken under consideration for publication only once the authors have clarified the details of the model they used.

Author Response

Point 1: In particular, the authors should clarify much bettter the meaning of the strategies as defined in Subsec. 2.3 (page 4 of 16). Why the strategy (for r=3) cddcdcc means cooperation in the first round, unconditional defection at the second round, and defection at the third round only if the opponent defected in the first and cooperated in the second round? With such notation, how can one distinguish the unconditional moves from the ones depending on the opponent's choises, and where are the opponent's choises indicated? The definition of the TfTx strategies is clearer, but in this case it would be good to explain why adding this peculiar type of strategies: why does utilizing that help us to understand better the behaviour of the model?

Response 1: We agree that the explanation could be much improved, and thank the reviewr for pointing out specific problems. To improve, we have moved the section on strategies forward to be the first subsection (2.1), first explained the TfTx strategies, indcluding our motivation to strat from these strategy sets (basically, to connect with the existing literature), then introduced the more difficult, complete sets of strategies where we rephrased the explanation how we code these. We hope that the resulting overhaul is now clear.

Point 2: Moreover, I would suggest to anticipate this Subsection (2.3 Strategies) at the end of the introductory paragraph of section 2, before the Subsec. 2.1 (Analysis), in order that the reader have a clear idea of the structure of the game when introduced to the analysis and statistics.

Response: as mentioned above, we have moved the entire description of strategies forward, so that Analysis and Simulations come after the explanation of the game strategies.

Point 3: More in general, there are many parameters and quantities which define the model (p, u, f, w...), so it is easy for a non-expert reader to get lost.

Response 3: to facilitate the non-expert reader, we have now introduced a new table 1 with the parameters defining the model.

Point 4: Finally, why adding the quantity K? Is this background fitness really necessary to the model? If yes, please justify more deeply its introduction.

Response 4: thanks for pointing this out, we have now justified this by pointing out that the model is more general that way because it covers a wider set of conditions for the evolution of cooperation in the PD game.

Point 5: The rest of the paper would result more understandable and easy to read once the definition of the model has been rewritten more clearly.

Response 5: We have also checked the text of results and discussion for clarity, and we trust that the revised version of our MS is now understandable.

Point 6: My last concern is the definition of the direct-undirect effects, being the first ones related to the interactions with the "founders" (the first to have a certain mutation, if I have understood correctly), and the second ones with its descendants. Hpw is it related to direct-undirect reciprocity (I reciprocate the action the opponent adopts with me withhim/herself, or with other agents)? This also need clarifications.

Response 6: Indeed, there is no relation between our direct-indirect mutation effect (which the reviewer understood correctly) and the direct-indirect reciprocity literature, as all our effects of mutation occur in the PD game which deals with direct reciprocity. Indirect effects of mutation are effects in subsequent generations of mutants that change the population structure, which is a very different phenomenon from indirect reciprocity which deals with game play between players that only meet once but have knowledge of previous actions of the opponent against other players.

Point 7: I repute that this paper can be taken under consideration for publication only once the authors have clarified the details of the model they used.

Response 7:  we trust that our revision has clarified the details of the model.

Reviewer 2 Report

I do not find enough value in this paper to warrant publication. 

I might have overlooked something, but it is not clear to me what we can learn from results. Maybe this is also because the paper is not well placed in the literature on the evolution of cooperation.

Author Response

Point 1: I do not find enough value in this paper to warrant publication. 

I might have overlooked something, but it is not clear to me what we can learn from results. Maybe this is also because the paper is not well placed in the literature on the evolution of cooperation.

Response 1: We are afraid we cannot use these unspecified comments for improvement of our MS, and note that they are in stark contrast to the comments of the other two reviewers who did elaborate on their points. Reviewer 1 judged our MS includes all relevant references, and reviewer 3 only suggested a few papers on collaboration to add. Also, we have spelled out what can be learned from these results, and the comments of the other two reviewers indicate that whereas the explanatino of the model could be improved, the take-home message of the work is very clear.

Reviewer 3 Report

The paper uses simulations to study the direct and indirect (due to the descendants of mutants) effect of mutations on the evolution of cooperation in finitely repeated prisoner's dilemmas. That they find that the latter effect is more important and that All-D is rather resilient, is, perhaps, to be expected, but the details of the dynamics are still interesting.

Comments:

It has been remarked in a recent survey (Newton, 2018 - "Evolutionary game theory: a renaissance", Games) that all models of the evolution of cooperation in PDs rely on inducing assortativity in the rates at which strategies C and D are played against one another. The authors should note this and give the citation.

Line 146, page 4. In your list of triplets, I think the second one should be DDC and not DDD.

Line 250, page 6. I think that {T,S,R,K} should be {T,S,R,P}.

Given the relative difficulty, even artificiality, of inducing cooperation in PDs, the authors may also wish to take a look at the recent literature on collaboration (a more general approach to cooperation), in, e.g.

Angus & Newton (2015), "Emergence of shared intentionality is coupled to the advance of cumulative culture." PLOS - Computational Biology.

Newton (2017), "Shared intentions: the evolution of collaboration." Games & Economic Behavior.

Rusch (2017), "Shared intentions: collaboration evolving." Working Paper. https://www.econstor.eu/handle/10419/174335

Author Response

Point 1: It has been remarked in a recent survey (Newton, 2018 - "Evolutionary game theory: a renaissance", Games) that all models of the evolution of cooperation in PDs rely on inducing assortativity in the rates at which strategies C and D are played against one another. The authors should note this and give the citation.

Response 1: thank you for pointing out this reference; we now use it in our MS (section Conclusions).

Point 2: Line 146, page 4. In your list of triplets, I think the second one should be DDC and not DDD.

Response 2: thank you for pointing out this careless typo - we have corrected the entire set of action sequences in that line.

Point 3: Line 250, page 6. I think that {T,S,R,K} should be {T,S,R,P}.

Response 3: Indeed, this is not the case (we vary parameter P at that point in the analysis) but we see that the notation can be confusing so in the revision we took K out of the curly brackets (it is not a parameter of the same type as the pay-offs T, S, R).

Point 4: Given the relative difficulty, even artificiality, of inducing cooperation in PDs, the authors may also wish to take a look at the recent literature on collaboration (a more general approach to cooperation), in, e.g.

Angus & Newton (2015), "Emergence of shared intentionality is coupled to the advance of cumulative culture." PLOS - Computational Biology.

Newton (2017), "Shared intentions: the evolution of collaboration." Games & Economic Behavior.

Rusch (2017), "Shared intentions: collaboration evolving." Working Paper. https://www.econstor.eu/handle/10419/174335

Response 4: Thanks for pointing us to this useful literature, we have now incorporated this in the revised MS (section Conclusions). (We have not included the citation to the working paper, as it is not customary in Biology to cite papers that are not yet in press in a peer-reviewed journal.)

Round  2

Reviewer 1 Report

After the revisions carried out by the authors, the manuscript is now good enough to be published in Games.

Reviewer 2 Report

The authors clarified what they find, but in my opinion the authors did not clarify what is the added value of such findings - in terms of our understanding of the evolution of cooperation.

In other words, I do not see what new insight we get.